# Detection of Alpha- and Betacoronaviruses in *Miniopterus fuliginosus* and *Rousettus leschenaultii*, two species of Sri Lankan Bats

**DOI:** 10.3390/vaccines9060650

**Published:** 2021-06-15

**Authors:** Therese Muzeniek, Thejanee Perera, Sahan Siriwardana, Dilara Bas, Fatimanur Kaplan, Mizgin Öruc, Beate Becker-Ziaja, Franziska Schwarz, Gayani Premawansa, Sunil Premawansa, Inoka Perera, Wipula Yapa, Andreas Nitsche, Claudia Kohl

**Affiliations:** 1Robert Koch Institute, Centre for Biological Threats and Special Pathogens, Highly Pathogenic Viruses (ZBS 1), 13353 Berlin, Germany; muzeniekt@rki.de (T.M.); BasD@rki.de (D.B.); KaplanF@rki.de (F.K.); OerucM@rki.de (M.Ö.); SchwarzF@rki.de (F.S.); NitscheA@rki.de (A.N.); 2Institute of Biochemistry, Molecular Biology and Biotechnology, University of Colombo, Colombo 00300, Sri Lanka; Thejanee90@gmail.com; 3IDEA (Identification of Emerging Agents) Laboratory, Department of Zoology and Environment Sciences, University of Colombo, Colombo 00300, Sri Lanka; sahan@zoology.cmb.ac.lk (S.S.); suviprema@gmail.com (S.P.); icperera@sci.cmb.ac.lk (I.P.); wipula@gmail.com (W.Y.); 4Robert Koch Institute, Centre for International Health Protection, Public Health Laboratory Support (ZIG 4), 13353 Berlin, Germany; Becker-ZiajaB@rki.de; 5Colombo North Teaching Hospital, Ragama 11010, Sri Lanka; gavisprema@gmail.com

**Keywords:** bat coronavirus, *Miniopterus fuliginosus*, *Rousettus leschenaultii*, Sri Lanka, cave-dwelling, sympatric colony, alphacoronavirus, betacoronavirus

## Abstract

Bats are known to be potential reservoirs of numerous human-pathogenic viruses. They have been identified as natural hosts for coronaviruses, causing Severe Acute Respiratory Syndrome (SARS) in humans. Since the emergence of SARS-CoV-2 in 2019 interest in the prevalence of coronaviruses in bats was newly raised. In this study we investigated different bat species living in a sympatric colony in the Wavul Galge cave (Koslanda, Sri Lanka). In three field sessions (in 2018 and 2019), 395 bats were captured (*Miniopterus, Rousettus, Hipposideros* and *Rhinolophus* spp.) and either rectal swabs or fecal samples were collected. From these overall 396 rectal swab and fecal samples, the screening for coronaviruses with nested PCR resulted in 33 positive samples, 31 of which originated from *Miniopterus fuliginosus* and two from *Rousettus leschenaultii*. Sanger sequencing and phylogenetic analysis of the obtained 384-nt fragment of the RNA-dependent RNA polymerase revealed that the examined *M. fuliginosus* bats excrete alphacoronaviruses and the examined *R. leschenaultii* bats excrete betacoronaviruses. Despite the sympatric roosting habitat, the coronaviruses showed host specificity and seemed to be limited to one species. Our results represent an important basis to better understand the prevalence of coronaviruses in Sri Lankan bats and may provide a basis for pursuing studies on particular bat species of interest.

## 1. Introduction

Bats (*Chiroptera*) are an order of mammals with the highest variety of species worldwide [1]. In Sri Lanka, the suborders of *Yinpterochiroptera* and *Yangochiroptera* are both represented in high numbers, accounting for almost 1/3 of the Sri Lankan mammals with 30 different species [2]. Their species variety and other unique features, specifically their ability to fly long distances, their migratory behavior and gregarious roosting habits, make them important virus reservoirs [1]. The study of these unique animal hosts is an important field of virology in order to learn more about the co-evolution of bats and viruses and the potential zoonotic transmission of viruses from bats to humans. Furthermore, an extensive knowledge about viral transmission mechanisms and virus-host interactions facilitates the development of antiviral drugs and vaccines. In the past decades, a number of zoonotic events has been resulting in an increase of emerging infectious diseases. Outbreaks of Ebola and Marburg viruses, Hendra virus, Nipah virus and coronaviruses (SARS-CoV and SARS-CoV-2, MERS-CoV) are among the most prominent examples [3,4,5]. Coronaviruses (CoV) are a family of RNA viruses that can be subdivided into the genera of alpha- (α-CoV), beta- (β-CoV), gamma- (γ-CoV) and delta-coronaviruses (δ-CoV) [6]. So far, predominantly α-CoVs and β-CoVs were detected in bat species which may be their natural reservoir, although they can also be found in other domestic animals such as swine and horses or in wild animals such as donkeys [7]. Most probably, γ-CoVs and δ-CoVs derived from bird CoVs. Different CoVs such as HCoV-NL63 and HCoV-229E (α-CoV), HCoV-OC43 and HKU1 (β-CoV) cause mild respiratory symptoms in humans [8]. The group of β-CoVs also includes virus species which can cause severe respiratory symptoms and which have the potential to spread rapidly and easily among humans, demonstrated in the past and current pandemics of SARS-CoV, MERS-CoV and SARS-CoV-2. Especially under these circumstances, the understanding of bats and their role as reservoirs of certain viruses has become of special interest and is being enhanced in numerous Asian countries [9,10,11]. Sri Lanka has a rich biodiversity, also reflected by the presence of 30 different bat species inhabiting the island [12]. With such a high diversity, it can be assumed that α- and β-CoVs would be present in Sri Lankan bat populations, as bats are major hosts of these viruses. Already in 2018, Kudagammana et al. have detected CoV in flying foxes (*Pteropus medius*) in Sri Lanka [13]. Our research was focused on a population of bats in the Wavul Galge cave (Koslanda, Sri Lanka), one of the largest natural caves in Sri Lanka which is permanently occupied by five species of bats. With this study we expand the evidence of α- and β-CoVs in the two cave-dwelling Sri Lankan bat species *R. leschenaultii* and *M. fuliginosus*.

## 2. Materials and Methods

Investigative research on Sri Lankan bats was approved by the local governmental authority (Department of Wildlife Conservation, Sri Lanka) and conducted in accordance with relevant guidelines and regulations. Samples from Sri Lankan cave-dwelling bats roosting in the Wavul Galge cave were taken at three different points in time (March and July 2018 and January 2019). A total of 395 bats belonging to the genera *Miniopterus*, *Rousettus*, *Rhinolophus* or *Hipposideros* were captured and kept in bat holding bags until further processing to avoid sampling one individual twice. Sampling of the bats was performed while using adequate personal protection equipment, namely safety gloves, safety glasses and FFP3 masks. The bat species was determined macroscopically and documented. Additionally, oral swabs were taken from each bat for molecular species identification based on the cytochrome B gene [14]. Either rectal swabs were taken with sterile swabs, or feces was collected with forceps from the bat holding bags if droppings were available. Swabs and fecal samples were collected in tubes without any additives and stored natively in liquid nitrogen for transportation. For further processing, 500 µL of sterile PBS were added to rectal swabs or fecal samples. Rectal swab samples were mixed by vortexing, and fecal samples were homogenized by using ceramic beads and the FastPrep-24 device (MP Biomedicals, Eschwege, Germany). After a centrifugation step the supernatants were collected and used for RNA extraction with the Viral RNA Mini Kit (QIAGEN, Hilden, Germany).

Extracted RNA was transcribed to cDNA with a random hexamer primer by using SuperScript IV Reverse Transcriptase (Invitrogen, Carlsbad, CA, USA) according to the manufacturer’s instructions. The coronavirus screening was performed with a nested PCR assay [15] which is designed on the highly conserved RNA-dependent RNA polymerase (RdRP) gene of the coronavirus genome and amplifies a product of 455 bp. The original protocol was slightly adapted as follows, using cDNA instead of RNA as sample material and Platinum Taq DNA Polymerase (Invitrogen, Carlsbad, CA, USA) for PCR amplification. For the first round of the nested PCR assay, mixtures contained 300 nM of primer PC2S2 and 900 nM of primer PC2AS1, 2.5 mM MgCl_2_, 250 µM dNTPs, 1x Platinum Taq Buffer and 0.5 U of Platinum Taq DNA polymerase. Water was added to a final volume of 23 µL and 2 µL of cDNA were added per reaction. Positive and negative controls were included in each PCR set up to validate the results. The thermal cycling of the PCR was performed as described in the original protocol [15]. For the second round of the nested PCR assay, mixtures contained 300 nM of primer PCS and 400 nM of primer PCNAs; all other reagents were used in the same concentrations as in the first round. Water was added to a final volume of 23 µL and 2 µL of the first-round PCR product were added. Thermal cycling of the second PCR round was performed as described in the original protocol [15].

Products of both PCR rounds were run and analyzed simultaneously on a 1.5% agarose gel containing DNA Stain G (SERVA, Heidelberg, Germany). Positive PCR products were purified by using MSB Spin PCRapace Kit (Invitrogen, Carlsbad, CA, USA) and sequenced with a BigDye Terminator Cycle Sequencing Kit on an Applied Biosystems 3500 Dx Genetic Analyzer, using the corresponding forward and reverse primers for each strand.

Sanger sequences were analyzed by using Geneious Prime software, and low-quality bases at the end of each sequence were trimmed before further processing. A nucleotide alignment of 384 nt was calculated by using MAFFT algorithm v7.450 [16]; the alignment contained all sequences of the positive samples as well as CoV reference strains obtained from the NCBI database. Editing was performed with MEGA7. A phylogenetic tree was calculated by using MrBayes version 3.2.6 [17]. The model HKY85 with gamma-distributed rate variation was selected for these calculations; parameters were set as follows: number of runs: four; number of generations: 10,000,000; subsampling frequency: 10 and burn-in: 50%. The reference strain avian infectious bronchitis virus (NC_001451, ICTV type species for γ-CoVs) was selected as outgroup for the calculations. The phylogenetic tree was visualized with the Geneious Prime software.

## 3. Results

In total, 255 rectal swabs and 141 fecal samples from different bat species were tested by using the adapted nested PCR protocol for the generic detection of coronaviruses (Table 1).

A total of 33 samples were tested positive (Table 2). The positive samples of this study were named after the corresponding bat species, the sampling session dates (March 2018 = 03-18, July 2018 = 07-18, January 2019 = 01-19) and the internal sample number. For example, the positive rectal swab (RS) sample RS170 from a *M. fuliginosus* bat collected in July 2018 was named batCoV/MinFul/07-18/RS170.

Sanger sequencing of the positive samples revealed a consensus sequence of at least 384 nucleotides. The sequences of batCoV/MinFul/07-18/RS91 and batCoV/MinFul/07-18/RS94 showed 100% identity, as did the sequences batCoV/MinFul/07-18/RS114 and batCoV/MinFul/01-19/F356. All other samples were unique in the 384 nt sequence with at least one nucleotide difference to the others (with identities ranging from 65% to 99%). This is visualized by an alignment-based nucleotide heatmap of all Sri Lankan bat samples (Figure 1). Twenty-seven of the *M. fuliginosus* sequences (both rectal swabs and fecal samples) have a high identity of 100–95%. Further four sequences have an identity of less than 80–75% to the other samples, while three of them (batCoV/MinFul/07-18/F153, -01-19/F347 and -01-19/F353) share a high identity of 98–99% among each other. Only the sample batCoV/MinFul/07-18/F350 has a lower identity of 77% to the three sequences and the lowest identity of 75–76% to all other *M. fuliginosus* sequences.

Apart from this, the two positive samples from *R. leschenaultii* bats share an identity of 56–63% to the *M. fuliginosus* sequences. Among each other, the sequence identity is 90%.

Furthermore, a phylogenetic tree was calculated with all sequences obtained from this study and other CoV reference strains from the NCBI database (Figure 2). In general, all *M. fuliginosus* sequences were allocated to the branch of α-CoV. The 29 highly identical sequences (95–100% identity) form a separate phylogenetic clade with three further *Miniopterus* spp. sequences from China and Hongkong [18,19]. The four *M. fuliginosus* sequences showing a lower identity on nucleotide level (Figure 1) were allocated to different groups in the phylogenetic tree. While three of them cluster with a *Miniopterus* spp. batCoV HKU8 strain, the least identical sequence (batCoV/MinFul/07-18/F350) forms a common clade with a *Miniopterus* spp. batCoV HKU7 strain. Human CoV-like 229E and NL63 are assigned also to the α-CoVs but have a higher distance to bat CoV strains.

Both sequences from *R. leschenaultii* samples were allocated to the branch of β-CoV. Within this branch, both sequences form a small group with two other β-CoVs from *Rousettus* spp. sampled in India [20]. Other β-CoVs like HCoV OC43 and SARS-CoV-2 strains from China and Sri Lanka form separate groups and have a higher distance from this branch.

## 4. Discussion

In this study we detected α- and β-CoVs in *R. leschenaultii* and *M. fuliginosus* bats in Sri Lanka for the first time. Only in 2018, another study found β-CoVs in Sri Lankan flying foxes (*Pteropus medius*), but since there was no access to the sequences we were not able to include them in our phylogenetic analysis for comparison [13].

### 4.1. α-CoVs in Miniopterus fuliginosus Bat Samples

In total, we found 31 α-CoVs from *M. fuliginosus* rectal swabs and fecal samples. The focus of the bat sampling in our study was on *M. fuliginosus* bats, mainly because of their migration behavior. Therefore, we collected the highest number of samples from this species during all three sampling sessions. In return, most of our samples that were screened positive were obtained from *M. fuliginosus* bats. A first comparison in a nucleotide alignment and the corresponding heatmap shows that the sequences have a high variety among each other. Even the first cluster of 29 samples partially reveals identities of 97% and less which can be considered low for this short sequence on the highly conserved RdRP gene. The structural functionality of the RdRP gene is essential for viral replication; this gene is therefore less susceptible to mutation events than other parts of the genome [21]. Despite this, RNA viruses like CoVs replicate with a mutation rate of one per 1000 to 10,000 nucleotides [22], which results in the development of novel coronaviruses and also explains a high diversity of the RdRP gene on nucleotide level. The separation of the *M. fuliginosus* sequences into at least two clusters therefore indicates the presence of different α-CoVs within the species. The phylogenetic analysis supports the assumption that different viral strains are present which all belong to the group 1 α-CoVs [22]. The related CoV HKU8 and CoV HKU7 strains in that cluster were obtained from *Miniopterus* spp. as well. As reported elsewhere before, different bat species from the genus *Miniopterus* probably serve as hosts for HKU7, HKU8 and closely related CoVs [18]. The presence of multiple viral strains within the *M. fuliginosus* cave population would be reasonable, considering the fact that these bats migrate seasonally from small surrounding colonies to the Wavul Galge cave, using it as a pre-maternity cave [2]. The migration time is between July and August and matches our second sampling session (July 2018), when we found most of the positive samples (25 α-CoVs in *M. fuliginosus* bats, see Table 1) [12]. Another explanation might also be that 65% of the positive samples were from female *M. fuliginosus* bats. In general, the number of female bats was increased at that sampling point, as 85% of the sampled *M. fuliginosus* bats in July 2018 were females. Persistence and circulation of different α-CoV strains in the small neighboring colonies could be assumed, and further studies would be necessary to investigate whether transmission or exchange of these different virus strains within the *M. fuliginosus* species occurs when they migrate to the Wavul Galge cave.

### 4.2. β-CoVs in Rousettus leschenaultii Bat Samples

In addition to the detected α-CoVs in *M. fuliginosus* samples, we found two β-CoVs in *R. leschenaultii* bat samples. *R. leschenaultii* is a fruit bat and the only cave-dwelling megachiropteran species [12]. In contrast to the *M. fuliginosus* species, they do not show seasonal migration behavior but are long-term inhabitants of the Wavul Galge cave. Both positive samples were taken during the same session (July 2018); the low sequence identity of 90% suggests that we detected two different β-CoVs strains that probably persist in the *R. leschenaultii* population of the Wavul Galge cave. A follow-up study with a higher number of samples from *R. leschenaultii* bats could prove whether the virus can be found in more *R. leschenaultii* bats or whether even more β-CoVs strains are present in this species.

The phylogeny of both sequences allocates them to other β-CoVs of the HKU9 strain. *Rousettus* spp. in India [20], China [23,24] and Singapore [25] already tested positive for HKU9 β-CoVs and may be their natural reservoir.

### 4.3. Presumed Host Specificity of Bat CoVs

Apart from the presented results, the sampled *Rhinolophus* spp. and *Hipposideros* spp. were not tested positive for any CoVs. Generally speaking, the number of sampled bats was probably insufficient to make a final statement, and further sampling with an increased number of bats should follow to study the prevalence of CoV in these species more thoroughly. Another aspect is the shedding of viruses in some bat species, possibly influenced by seasonal changes and environmental conditions [26,27]. We collected samples in January, March and July. Sampling *Rhinolophus* spp. and *Hipposideros* spp. at other points in time might clarify whether these species shed CoVs in other seasons of the year. Interestingly, the prevalence of CoV in the Wavul Galge cave seems to be very host specific, although *M. fuliginosus*, *R. leschenaultii*, *Rhinolophus* spp. and *Hipposideros* spp. roost sympatrically in the cave. For example, we could only detect α-CoVs in *M. fuliginosus* bats, although *Hipposideros* spp. are also known to carry α-CoVs and are assumed to be the natural host of human CoV 229E [28]. Furthermore, *Rhinolophus* spp. are known to carry SARS-like β-CoVs and may be the natural reservoir of the pandemic SARS-CoV-2 [10]. Although transmission of CoVs is generally possible [10,29], it did not seem to occur in this sympatric colony. One reason may be the spatial separation of the different species inside the Wavul Galge cave; therefore, direct contact between the species is only likely when entering or exiting the cave. Still, the probability of aerosol-based transmission between the species in the cave should be considered to be very high. Therefore, long-term monitoring of all bat species in the cave could help understand host specificity and transmission dynamics of these viruses.

### 4.4. Evaluating the Risk of Viral Spillover to Humans

As a result of the Covid-19 pandemic, in 2021, Grange et al. developed an open-source risk ranking tool to evaluate the risk of viral spillover to humans and the spreading potential of these viruses [30]. For their ranking, they selected innovative risk factors, divided into host risk factors, environmental risk factors and virus risk factors. With the help of this tool (https://spillover.global/ranking-comparison; accessed on 10 June 2021), we evaluated the spillover risk of the α-CoVs and β-CoVs we detected in *R. leschenaultii* and *M. fuliginosus* samples in Sri Lanka. For this purpose, we used the phylogenetic tree and selected the closest related strains. For *R. leschenaultii*, we checked the bat coronavirus HKU9 which has a high ranking score of 80 out of 155 and can be found in position 14 of the overall risk ranking of 887 viruses (https://spillover.global/virus/21; accessed on 10 June 2021). The risk factors of this *Rousettus* bat coronavirus HKU9 are for example the high host diversity (found in 11 bat species), the global distribution of the virus and the high interaction of wildlife, domestic animals and humans in these regions. For the α-CoVs from *M. fuliginosus* samples, we checked three of the closest related CoVs. The bat coronavirus 1 was the closest related strain for most of the α-CoVs from *M. fuliginosus* samples. With a risk score of 72 out of 155 it is ranked at position 40 (https://spillover.global/virus/49; accessed on 10 June 2021). The virus can be found in six different bat species, including three *Miniopterus* spp. and one species from the genera *Hipposideros*, *Myotis* and *Rhinolophus*, respectively. This host range may be of interest because *Hipposideros* and *Rhinolophus* bat species roost in the same cave, but were not tested positive for any CoVs so far. As discussed before, long-term monitoring may reveal CoVs in these species as well and could further support the host range given by the spillover ranking tool. Another important factor that has an impact on the spillover risk is the interaction of animals and humans in the region where the virus is found. The distribution of the virus is only semi-global and may explain its lower spillover risk. The Miniopterus bat coronavirus HKU8 can be found in position 107 with a spillover risk of 65 out of 155 (https://spillover.global/virus/114; accessed on 10 June 2021). It can only be found in *Miniopterus* spp. and is distributed semi-globally; in its distribution area, the interaction between humans and host animals is high. The same applies to the last bat coronavirus HKU7 which can be found semi-globally in two *Miniopterus* spp. and one *Taphozous* spp., while the interaction of the hosts and wild animals with humans is rated as medium. Consequently, this virus has the low position of 382 in the overall ranking with a risk score of 56 out of 155.

To sum up, this tool provides a good ranking system to evaluate the risk of spillover events to humans. The results confirm what has already been discussed before by assigning a rather low risk for zoonotic events to the α-CoVs detected in the *M. fuliginosus* species. In contrast, the β-CoVs from *R. leschenaultii* samples were ranked with a higher risk, which is reasonable considering the fact that other related β-CoVs from bat species are already known for their role in different spillover events.

The risk factors that are included in this ranking tool (virus, host and environmental factors) point out the high impact of the interaction of humans and wildlife which contributes significantly to the accumulation of zoonotic events.

## 5. Conclusions

With this study, we provide the first molecular biological analysis of different viruses in bat species roosting in the Wavul Galge cave, Sri Lanka. We detected α-CoVs in *M. fuliginosus* and β-CoVs in *R. leschenaultii* bats. Our results indicate that different virus strains are persistently present within both populations. Detailed gene and genome analysis of the existing CoVs as well as further studies with a focus on the other sympatric species (*Rhinolophus* spp. and *Hipposideros* spp.) might provide more insight into the prevalence, circulation or persistence of different CoV strains in this cave. A long-term study would be helpful to examine the seasonal shedding of the viruses and the impact of migration behavior of the different bat species. Our results indicate that the detected CoVs are host specific for the respective bat species and, despite the sympatric cohabitation in the Wavul Galge cave, an inter-species transmission was not observed. This supports the assumption that only particular bat species serve as natural reservoir for harmful human-pathogenic viruses like SARS-CoV, while spill-over events may occur because of environmental impact and the intrusion of humans to the living areas of the bat species [31]. Studying bats and monitoring their viruses as well as the respectful interaction with their natural habitats are both important factors to better understand and prevent zoonotic transmission from bats to humans. Being the largest cave in Sri Lanka with as many as five sympatric bat species, numbering over 100,000 individuals, the Wavul Galge cave provides an excellent natural site for long-term monitoring of bat-borne viruses in Sri Lanka. Our study emphasizes the need to periodically monitor all bat species and their viruses in this cave.

## Figures and Tables

**Figure 1 vaccines-09-00650-f001:**
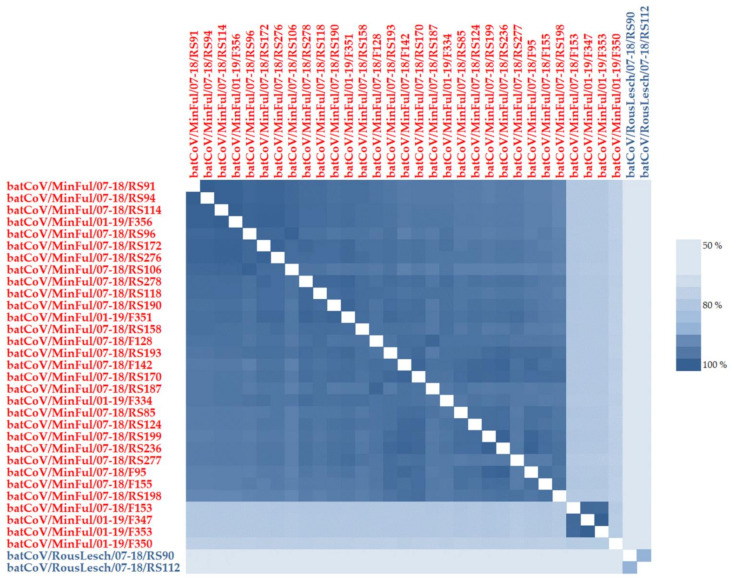
Heatmap based on a nucleotide alignment of 384 bp PCR product of the RdRP gene of the coronavirus genome. The figure illustrates the identities of all CoV-positive rectal swabs or fecal samples collected in the Wavul Galge cave, Koslanda, Sri Lanka, at three different points in time. Red sequences represent positive M. fuliginosus bats, whereas blue sequences indicate positive R. leschenaultii bats.

**Figure 2 vaccines-09-00650-f002:**
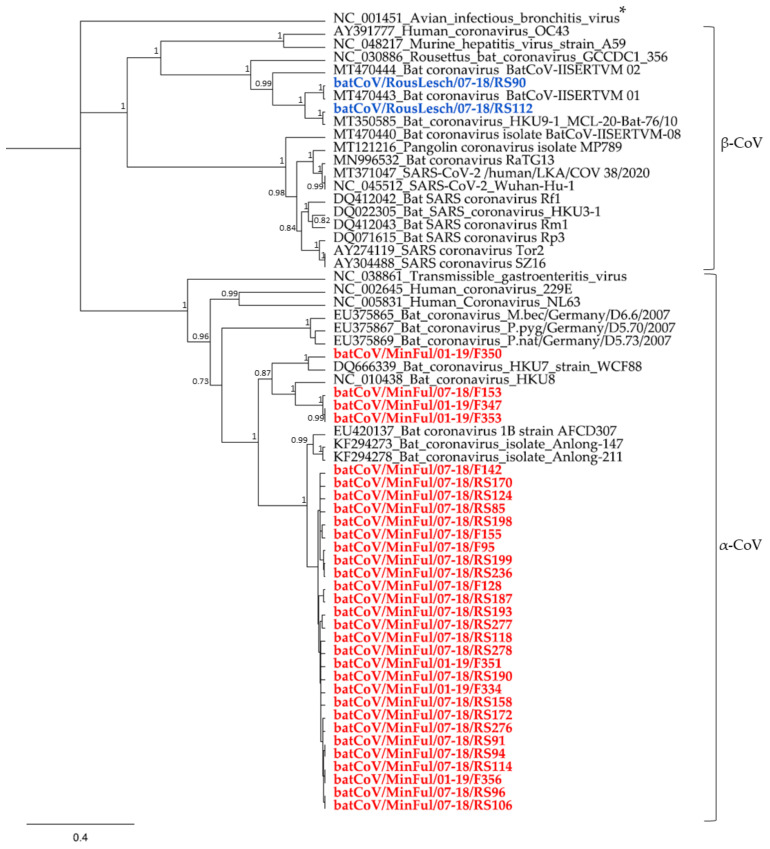
Phylogenetic tree of coronavirus sequences from Sri Lankan M. fuliginosus bats (given in red) and R. leschenaultii bats (given in blue) and other reference sequences of different α- and β-CoVs. The tree is divided into the two groups of α- and β-CoVs. In addition, the γ-CoV avian infectious bronchitis virus (NC_001451, marked with an asterisk) was included as outgroup for the calculation. The phylogenetic tree was calculated with Bayesian algorithm, and 5 mio trees were calculated with a subsampling frequency of 25 and a burn-in of 50%. Substitution model HKY85 was selected with a gamma-distributed rate variation. Branch label values for the 27 bat CoV sequences that form a joint clade are not displayed in the tree; their posterior probability values range between 0.52 and 1.

**Table 1 vaccines-09-00650-t001:** Overview of rectal swabs and fecal samples (CoV-positive/total of samples). Results are listed per bat genus and sampling session.

Genus	March 2018		June 2018		January 2019		Sampled Bats in Total
	Rectal swabs	Feces	Rectal swabs	Feces	Rectal swabs	Feces	
*Miniopterus*	0/3	0/0	20/115	5/76	0/4	6/27	31/225
*Rousettus*	0/8	0/2	2/11	0/0	0/16	0/3	2/40
*Hipposideros*	0/3	0/0	0/1	0/0	0/16	0/7	0/27
*Rhinolophus*	0/62	0/9	0/0	0/0	0/16	0/17	0/104
Total samples per session	76	11	127	76	52	54	

**Table 2 vaccines-09-00650-t002:** Details on the positive rectal swabs and fecal samples. The table lists the bat species determined by cytB sequencing (sequence data available on request), the sampling date, sex and forearm length of the sampled bats, the given name of the detected CoV and its GenBank accession number. All bats were captured and sampled in the Wavul Galge cave, Koslanda, Sri Lanka. n.a. = not applicable.

Sample	Species	Date	Sex	Forearm Length (cm)	Bat CoV Description	Accession Number
RS85	*M. fuliginosus*	07/07/18	m	4.76	batCoV/MinFul/07-18/RS85	MW987547
RS90	*R. leschenaultii*	07/07/18	f	5.67	batCoV/RousLesch/07-18/RS90	MW987539
RS91	*M. fuliginosus*	07/07/18	f	4.55	batCoV/MinFul/07-18/RS91	MW987548
RS94	*M. fuliginosus*	08/07/18	m	4.53	batCoV/MinFul/07-18/RS94	MW987549
RS96	*M. fuliginosus*	07/07/18	f	4.49	batCoV/MinFul/07-18/RS96	MW987554
RS106	*M. fuliginosus*	07/07/18	f	4.54	batCoV/MinFul/07-18/RS106	MW987555
RS112	*R. leschenaultii*	07/07/18	f	n.a.	batCoV/RousLesch/07-18/RS112	MW987540
RS114	*M. fuliginosus*	07/07/18	f	4.65	batCoV/MinFul/07-18/RS114	MW987550
RS118	*M. fuliginosus*	07/07/18	f	4.65	batCoV/MinFul/07-18/RS118	MW987556
RS124	*M. fuliginosus*	07/07/18	f	4.65	batCoV/MinFul/07-18/RS124	MW987546
RS158	*M. fuliginosus*	08/07/18	f	4.41	batCoV/MinFul/07-18/RS158	MW987566
RS170	*M. fuliginosus*	08/07/18	f	4.91	batCoV/MinFul/07-18/RS170	MW987545
RS172	*M. fuliginosus*	08/07/18	f	4.64	batCoV/MinFul/07-18/RS172	MW987552
RS187	*M. fuliginosus*	08/07/18	f	4.61	batCoV/MinFul/07-18/RS187	MW987563
RS190	*M. fuliginosus*	08/07/18	m	4.65	batCoV/MinFul/07-18/RS190	MW987559
RS193	*M. fuliginosus*	08/07/18	f	4.58	batCoV/MinFul/07-18/RS193	MW987560
RS198	*M. fuliginosus*	08/07/18	f	4.54	batCoV/MinFul/07-18/RS198	MW987564
RS199	*M. fuliginosus*	08/07/18	f	4.51	batCoV/MinFul/07-18/RS199	MW987542
RS236	*M. fuliginosus*	09/07/18	f	4.67	batCoV/MinFul/07-18/RS236	MW987543
RS276	*M. fuliginosus*	10/07/18	f	4.45	batCoV/MinFul/07-18/RS276	MW987553
RS277	*M. fuliginosus*	10/07/18	m	4.65	batCoV/MinFul/07-18/RS277	MW987561
RS278	*M. fuliginosus*	10/07/18	f	4.41	batCoV/MinFul/07-18/RS278	MW987557
F95	*M. fuliginosus*	07/07/18	f	4.38	batCoV/MinFul/07-18/F95	MW987541
F128	*M. fuliginosus*	07/07/18	m	4.55	batCoV/MinFul/07-18/F128	MW987562
F142	*M. fuliginosus*	07/07/18	f	4.69	batCoV/MinFul/07-18/F142	MW987544
F153	*M. fuliginosus*	07/07/18	m	4.59	batCoV/MinFul/07-18/F153	MW987568
F155	*M. fuliginosus*	08/07/18	f	n.a.	batCoV/MinFul/07-18/F155	MW987567
F334	*M. fuliginosus*	23/01/19	m	4.65	batCoV/MinFul/01-19/F334	MW987565
F347	*M. fuliginosus*	23/01/19	f	4.76	batCoV/MinFul/01-19/F347	MW987569
F350	*M. fuliginosus*	23/01/19	f	4.51	batCoV/MinFul/01-19/F350	MW987571
F351	*M. fuliginosus*	23/01/19	m	4.55	batCoV/MinFul/01-19/F351	MW987558
F353	*M. fuliginosus*	23/01/19	m	4.58	batCoV/MinFul/01-19/F353	MW987570
F356	*M. fuliginosus*	23/01/19	m	4.56	batCoV/MinFul/01-19/F356	MW987551

## Data Availability

The data presented in this study are openly available in GenBank (https://www.ncbi.nlm.nih.gov/genbank/), Accession numbers MW987539–MW987571.

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
