# Peer review of "Detection of Alpha- and Betacoronaviruses in Miniopterus fuliginosus and Rousettus leschenaultii, two species of Sri Lankan Bats"

_vaccines, 2021, doi:10.3390/vaccines9060650_

Round 1
Reviewer 1 Report
The paper "Detection of alpha- and betacoronaviruses in Miniopterus 2 fuliginosus and Rousettus leschenaultii, two species of Sri 3 Lankan bats" by Muzeniek and colleagues is an interesting addition to the body of knowledge of bat coronaviruses. The examination of multiple species occupying the same roost space and the species specific segregation of their coronaviruses are interesting points to ponder in assessing the spill over risk of these viruses.
The paper is generally well written with only some minor grammar errors (the use of "was" instead of were in a couple of places) and the coronavirus methods and results are robustly reported. The one piece of missing methodology was the cytB PCR and sequencing (I cannot see a reference to how this was done in the paper). Was this only done on positive samples or on all the samples in the study (if all it would be nice to have full species ID's rather than genus level IDs). These details need to be included (and sequences deposited) before the manuscript could be accepted for publication.
The sequence diversity within the miniopterus samples is interesting (and could perhaps be expanded with some discussion as to what you would expect for sequence diversity based on what has been previously reported in other species alphacoronaviruses. It could also be clearer in the phylogenetic tree which species of bat the various viruses have been derived from (eg it is not entirely clear in the manuscript that HKU7 and 8 were derived from miniopterus species and that is a useful piece of information for the implications of the results) .
I'm not sure that the addition of the virus spill over tool is entirely useful as I can't easily check the data underlying that ranking (for instance host species - is that number of bat species and which ones are they all bats from the same genus? ) It somewhat undermines your central argument that these are relatively species specific then presenting information from a curated database that says the opposite
Author Response
Response to Reviewer 1 Comments
Point 1: The paper "Detection of alpha- and betacoronaviruses in Miniopterus 2 fuliginosus and Rousettus leschenaultii, two species of Sri 3 Lankan bats" by Muzeniek and colleagues is an interesting addition to the body of knowledge of bat coronaviruses. The examination of multiple species occupying the same roost space and the species specific segregation of their coronaviruses are interesting points to ponder in assessing the spill over risk of these viruses.
Response 1: Thank you very much for your input to our manuscript, we adapted it according to your comments and hope to make some methods and results clearer to understand for the reader.
Point 2: The paper is generally well written with only some minor grammar errors (the use of "was" instead of were in a couple of places) and the coronavirus methods and results are robustly reported.
Response 2: Thank you for this comment, we went through the manuscript and corrected grammar errors, the changed version of the manuscript after revision was copy-edited thoroughly again.
Point 3: The one piece of missing methodology was the cytB PCR and sequencing (I cannot see a reference to how this was done in the paper). Was this only done on positive samples or on all the samples in the study (if all it would be nice to have full species ID's rather than genus level IDs). These details need to be included (and sequences deposited) before the manuscript could be accepted for publication.
Response 3: Thank you for your comment. We did the molecular cytB species determination on all sampled bats, we added that information in line 78. The full cytB (1,140 bp) gene of individual bats was sequenced and phylogentically analysed, this is subject of another study, published soon. In this study a short fragment of the cytB gene was analyzed to confirm the macroscopic species determination. This data is available on request. Therefore, only the species ID’s of all relevant positive bats is given in Table 2 and the general description of all positive and negative bats is given on genus level.
Point 4: The sequence diversity within the miniopterus samples is interesting (and could perhaps be expanded with some discussion as to what you would expect for sequence diversity based on what has been previously reported in other species alphacoronaviruses. It could also be clearer in the phylogenetic tree which species of bat the various viruses have been derived from (eg it is not entirely clear in the manuscript that HKU7 and 8 were derived from miniopterus species and that is a useful piece of information for the implications of the results).
Response 4: Thank you for that comment. Indeed, it is interesting that we found such a sequence diversity within the miniopterus samples. We discussed the mutation rate of CoV in general (line 200) and rephrased line 204 – 211. From most strains in the phylogenetic tree, the bat species was not given. For the relevant viruses like HKU7 and HKU we now describe
the bat species in the text to make clearer that all CoVs in the cluster were derived from Miniopterus species.
Point 5: I'm not sure that the addition of the virus spill over tool is entirely useful as I can't easily check the data underlying that ranking (for instance host species - is that number of bat species and which ones are they all bats from the same genus? ) It somewhat undermines your central argument that these are relatively species specific then presenting information from a curated database that says the opposite
Response 5: Thank you for your comment, we adapted that section accordingly (lines 274 – 291) and described the host species in detail to show that they are in some cases all from the same genus. We again pointed out that the presumed host specificity is only based on our current dataset and a long-term monitoring and more extensive sampling of the other bat species would be necessary and interesting.
Reviewer 2 Report
In the manuscript “Detection of alpha- and beta coronaviruses in Miniopterus fuliginosus and Rousettus leschenaultii, two species of Sri Lankan bats”, the authors intended to provide a basis for the screen of coronaviruses in bats species, using nested PCR. Afterwards a sanger sequencing and phylogenetic analysis was used for screening sequences similarity. In the end they have used an open source risk ranking tool to evaluate the risk of viral spillover to humans.
The Manuscript addresses an interesting and useful subject on a hot topic area nowadays, since it shows preliminary results on new coronaviruses variants. However, I believe “Vaccines” journal scope would require additional information on what these findings can contribute to the knowledge in the field, namely vaccine research, utilization and immunization and epidemiology. Additionally, I see several major methodological inaccuracies concerning this work, that are listed below:
-The first concern is related with sampling. What was made to guarantee that one animal is not caught up twice? A control should be implemented to avoid duplication on the counting of virus same variant. Ultimately this is important in phylogenetic analysis of the variants.
-The authors obtained some positive samples on swabs and on feces from the same bat. Correspondence on that samples wasn’t made, i.e., grouping the samples from the same animal. Also, from the results it was possible to observed that samples (feces and swabs) contained different Cov variants and none presented mixed populations. Nothing was stated about that, neither an explanation was given for different variants in the same animal.
-Some virus were excreted on feces but were negative in swabs (January 2019). This result is expected? Shouldn’t both samples be presumably positive? What kind of controls were provided?
-The authors didn’t refer the performance of any controls using nested PCR, so no certainty on the results must be claimed
-What is the reason for only 2 bat species have positivity to the presence of coronavirus?
Author Response
Response to Reviewer 2 Comments
Point 1: In the manuscript “Detection of alpha- and beta coronaviruses in Miniopterus fuliginosus and Rousettus leschenaultii, two species of Sri Lankan bats”, the authors intended to provide a basis for the screen of coronaviruses in bats species, using nested PCR. Afterwards a sanger sequencing and phylogenetic analysis was used for screening sequences similarity. In the end they have used an open source risk ranking tool to evaluate the risk of viral spillover to humans.
The Manuscript addresses an interesting and useful subject on a hot topic area nowadays, since it shows preliminary results on new coronaviruses variants.
Response 1: Thank you very much for your input to our manuscript, we adapted it according to your comments and hope to make some methods and results clearer to understand for the reader.
Point 2: However, I believe “Vaccines” journal scope would require additional information on what these findings can contribute to the knowledge in the field, namely vaccine research, utilization and immunization and epidemiology.
Response 2: Thank you for your comment. The content of our manuscript is rather focused on the journal’s special issue topic “Research in Bat-Borne Zoonotic Viruses”, were we intend to publish the manuscript. In the introduction (line 44) we now point out how research on bats and zoonotic viruses adds to vaccine research.
Point 3: The first concern is related with sampling. What was made to guarantee that one animal is not caught up twice? A control should be implemented to avoid duplication on the counting of virus same variant. Ultimately this is important in phylogenetic analysis of the variants.
Response 3: Thank you for this comment. We set up the bat sampling in front of the cave and placed the captured bats in bat holding bags. At the end of the catching process we sampled the bats one after another and released them afterwards. Sampling one bat twice was therefore not possible. We adapted the description of the sampling method (see line 75) to clarify.
Point 4: The authors obtained some positive samples on swabs and on feces from the same bat. Correspondence on that samples wasn’t made, i.e., grouping the samples from the same animal. Also, from the results it was possible to observed that samples (feces and swabs) contained different Cov variants and none presented mixed populations. Nothing was stated about that, neither an explanation was given for different variants in the same animal. Some virus were excreted on feces but were negative in swabs (January 2019). This result is expected? Shouldn’t both samples be presumably positive? What kind of controls were provided?
Response 4: Thank you for that comment. We did not collect feces and rectal swabs from the same bat individual. We collected either feces or, if feces was not available, we took a rectal
swab. Therefore there is only one positive or negative result per bat possible. As given in the result table 2, the individual sample numbers are all unique and represent an individual bat respectively. We changed the corresponding sentence in the method section (line 82) to make clearer that from each bat only feces or rectal swab was taken, but not both sample types for each bat.
Point 5: The authors didn’t refer the performance of any controls using nested PCR, so no certainty on the results must be claimed.
Response 5: Thank you for this comment, we added the information about the PCR controls in line 100-101
Point 6: What is the reason for only 2 bat species have positivity to the presence of coronavirus?
Response 6: Thank you for this comment, here we would like to refer to the discussion, section “Presumed host specificity of bat CoVs” (line 240 – 261), were we try to evaluate the possibility of viral transmission between the species inhabiting the cave on one hand, and a possible host specificity of these viruses on the other hand. With our data we cannot provide a definite answer, but we try to provide an outlook on how to answer this question.
Round 2
Reviewer 2 Report
After the authors' clarifications, the manuscript is much clearer,so I don´t have further questions.
Therefore, I recommend it to publication.
I had no prior information about the submission of the manuscript to the
special issue “Research in Bat-Borne Zoonotic Viruses”. However,
I understand the submission in this issue and in that case, I agree that it fits with the subject.